# Characteristics and Functional Application of Cellulose Fibers Extracted from Cow Dung Wastes

**DOI:** 10.3390/ma16020648

**Published:** 2023-01-09

**Authors:** Xiangjun Yang, Lu Li, Wuyun Zhao, Mengyang Wang, Wanxia Yang, Yuhang Tian, Ruizhe Zheng, Shuhang Deng, Yongsong Mu, Xiaodong Zhu

**Affiliations:** 1School of Mechanical and Electrical Engineering, Gansu Agricultural University, Lanzhou 730070, China; 2School of Mechanical Engineering, Chengdu University, Chengdu 610106, China; 3Huarui Agricultural Company, Zhangye 734500, China

**Keywords:** cow dung, cellulose, fiber, extraction, paper-making

## Abstract

The widespread use of petroleum-based products has led to increasing environmental and ecological problems, while the extraction and application of various natural cellulose fibers have received increasing attention. This research focuses on the extraction of cellulose fibers from cow dung using different treatments: hot water, hydrogen peroxide (H_2_O_2_), sodium hydroxide (NaOH) and potassium hydroxide (KOH) boilings, as well as a selection of the best quality cow dung fibers for papermaking with quality control. The study’s objective is to find a sustainable method to extract as much material as possible from renewable biomass feedstock. The results show that the best extraction rate is obtained by KOH boiling with 42% cellulose fibers extracted. Corresponding handmade paper has a burst index of 2.48 KPam^2^/g, a tear index of 4.83 mNm^2^/g and a tensile index of 26.72 Nm/g. This project expands the sources of natural cellulose fibers to an eco-friendly and sustainable one and opens up new applications for cow dung.

## 1. Introduction

Plastics derived from non-renewable petroleum resources contribute to environmental pollution because of their inability to biodegrade. That has stimulated research towards the development of alternative biodegradable materials that can replace some plastic products’ applications in nutrition bowls, packaging bags, mulching film, etc. [1]. Lignocellulose-based materials receive much attention because of their numerous advantages such as lightweight, cost-effectiveness, non-toxicity, high mechanical strength, biodegradability and biocompatibility. Among them, cellulose fiber is one of the main components of natural fibers found in wood (softwood and hardwood [2]) and plants (banana [3], palm tree [4], hemp [5], coconut husk [6], flax [7], cotton [8], and sisal [9]). Cellulose fiber is a linear, high-molecular-weight homopolysaccharide made up of β-1,4-D-glucose units [10]. Its three free hydroxyl groups per monomer unit can form strong hydrogen bonds, which give it a very cohesive nature and a microfibrillated structure [11]. Extracted cellulose fibers can produce various cellulose products, which have been used in industrial production, packaging, and many types of materials in infrastructure construction [12,13].

The process of cellulose fiber extraction was mostly focused on the usage of natural plants such as seaweed [14], Phormium tenax [15], tomato plant residue [16] and Eleusine indica grass [17]. Cellulose fibers can also be extracted from waste biomass, such as apple pomace, mulberry leaves, cassava pomace, bagasse, corn straw, and peanut shell, etc. [18,19]. However, the reuse of cellulose fibers in manure from animal husbandry, particularly the most prevalent cow dung, has seldomly been researched. The cows, as ruminants, eat plant-based food, which is taken into the cows’ four-chambered stomach, where it is digested and broken down into smaller pieces before the excretion of dung [20]. Within these chambers, the food is subjected to mechanized processing while different digestive enzymes and microorganisms break down the plant fibers, notably hemicellulose and pectin. However, the fiber digestion process is not complete and a significant quantity of cellulose fiber remains in cow dung [21].

Cow dung, as a waste of animal husbandry, is another primary pollution in rural areas and has become an urgent problem. For example, in Zhangye City in the northwest of China, animal husbandry is becoming a booming enterprise thanks to its about 600,000 heads of cattle. While beneficial to the local economy, animal husbandry generates a waste disposal problem with approximately 6.48 × 10^6^ tons of cow dung per year, which pose a serious public health issue and environmental pollution. So, it is important to research the efficient uses of animal waste biomass. A lot of research has been conducted on the utilization of cow dung. For example, cow dung is used as compost to keep the soil healthy and increase crop yields [22], as a base for vermicomposting [23], or as biochar to prevent soil nutrient losing and increase crop yields [24]. In addition, cow dung can be found in widespread applications, including clean energy [25,26], sewage treatment [25,26], and reinforcement materials in composite materials [27]. However, extracting cellulose fibers from cow dung, characterizing them, and employing them in functional applications, has not been yet extensively explored.

In this work, as much cellulose fibers as possible were extracted from the renewable biomass feedstock that is cow dung by four treatments: hot water, H_2_O_2_, NaOH, and KOH boilings. The optimum extraction treatment was identified by testing the characteristics of cellulose fibers from different treatments, while respecting environmental and sustainable development. The application of extracted cellulose fibers in forming handmade paper was preliminarily explored. Scanning electron microscopy (SEM), X-ray diffraction (XRD), Fourier transform infrared spectroscopy (FTIR), and Thermogravimetric analysis (TGA) were used to characterize the extracts from cow dung. Alkali (KOH and NaOH) extractions, which effectively eliminate residual lignin and hemicellulose from cow dung, gave much better extraction rates than H_2_O_2_ and water boilings. In addition, the residual black liquid with KOH extraction could be diluted and used as a liquid fertilizer [28]. This project expands the sources of natural cellulose fibers to an eco-friendly and sustainable one and opens up new applications for cow dung.

## 2. Experimental Procedure

### 2.1. Materials

The cow dung was obtained from HuaRui Agriculture Co., Ltd., Zhangye City, Gansu Province, China. The cow dung fibers were initially cleaned and dried to remove the sands included. The Analytical grade chemical reagents such as KOH, NaOH, and H_2_O_2_ (30%) in the experiment were purchased from Chengdu Kelong Chemical Co., Ltd., Chengdu, China.

### 2.2. Experimental Process

The cow dung was stirred and cleaned until the pH value of the aqueous solution was neutral. The obtained cow dung fibers were dried at 80 °C for 12 h and stabilized in a dryer for 24 h, and the corresponding sample is named S1. Four different treatments (Hot water, H_2_O_2_, NaOH, and Sodium KOH boiling extractions) were used to extract cellulose fibers from crude fiber, and the specific treatment process is shown in Figure 1.

#### 2.2.1. Hot Water Extraction (S2)

The quantitative crude fibers (5 g) were taken in deionized water and kept at 90 °C for 2 h, washed to neutrality, dried at 80° for 12 h and stabilized in a dryer for 24 h. Sample is named S2.

#### 2.2.2. Hydrogen Peroxide Extraction (S3)

5 g of crude fiber was treated with 5 wt.% H_2_O_2_ solution at 90 °C with stirring for 2 h, washed to neutrality, dried at 80° for 12 h and stabilized in a dryer for 24 h. Sample is named S3.

#### 2.2.3. Sodium Hydroxide Extraction (S4)

5 g of crude fiber was treated with 5 wt.% NaOH solution at 90 °C with stirring for 2 h, washed to neutrality, dried at 80° for 12 h and stabilized in a dryer for 24 h. Sample is named S4.

#### 2.2.4. Potassium Hydroxide Extraction (S5)

5 g of crude fiber was treated with 5 wt.% KOH solution at 90 °C with stirring for 2 h, washed to neutrality, dried at 80° for 12 h and stabilized in a dryer for 24 h. Sample is named S5.

### 2.3. Characterization

#### Composition Analysis

The chemical compositions of cow dung were carried out according to standardized methods. The water content was determined following GB 02677.2-2011-T, the ash content by GB 02677.3-1993-T, and the water extract content under GB 02677.4-1993-T. The 1% sodium hydroxide extract content was determined by GB 02677.5-1993-T. The organic solvent extract content follows GB 02677.6-1994-T. The measurements of acid-insoluble contents of all samples were repeated at least twice, and the difference in collected data values was less than 5% within the experimental error.

Fourier transform infrared spectroscopy (FTIR)

The Fourier transform infrared (FT-IR) spectra of all samples were recorded using a PerkinElmer Spectrum 100 spectrometer equipped with ZnSe single-reflection ATR accessory in the wavenumber range of 4000~400 cm^−1^ at the resolution of 2 cm^−1^ and 32-times scanning in air [29].

X-ray diffraction (XRD)

Phase structure of samples was determined by XRD at 40 kV, 30 mA with monochromatic Cu-Kα radiation, typically with a scan speed of 0.5°/min and sampling pitch of 0.03° in a 2θ scale region of 10–90° (D2 PHASER diffractometer, BRUKER, Billerica, MA, USA) [30]. crystallinity index (CrI) of the samples were determined according to the empirical formula:(1)$$\mathrm{CrI}=\frac{\left(I_{200}-I_{\mathrm{am}}\right)}{I_{200}}\times 100\%$$
where I_200_ is the maximum diffraction intensity at the (200) plane, and I_am_ is the intensity diffraction at approximately 2θ = 15°.

Thermogravimetric analysis (TGA)

The thermal stability of all samples was evaluated by TGA (Mettler Toledo TGA, DSC 3+). The sample was heated from 30 °C to 800 °C at a heating rate of 10 °C/min under a nitrogen atmosphere (flow rate 25 mL/min).

Scanning electron microscopy (SEM)

The morphological characterizations of all samples were examined by scanning electron microscope (SEM, FEI Inspect F50, Hillsborough, OR, USA), and all samples were treated with gold spraying before detection.

### 2.4. Preparation and Testing of Cow Dung Paper

#### 2.4.1. Preparation of Cow Dung Paper

The diluted pulp suspension concentration is 2 g/L. The prepared hand sheets exhibited a base weight of 60 g/m^2^, according to ISO 5269-2. Handmade papers were conditioned at 23 °C and 50% humidity for 48 h, according to ISO 187.

#### 2.4.2. Testing of Cow Dung Paper

All the hand sheets tests were based on the following standards: ISO536, ISO 534, ISO 5636-3, ISO 1924-3, ISO 2758, and ISO 1974, to measure the basis weight, thickness, bulk, and permeability as well as the tensile, burst, and tear strengths.

## 3. Results and Discussions

### 3.1. Characterization of Cow Dung

#### 3.1.1. Chemical Composition

Table 1 shows the chemical analysis of cow dung and a variety of lignocellulosic plants. The results obtained from cow dung were compared with those from other raw materials reported in the literature. The results show that the cellulose and holocellulose contents are close to that of softwood, and lignin content is similar to those of hardwood [2]. The ash content is equivalent to the one of bagasse [18], and the NaOH extract is slightly higher than the one of cotton cellulose [31]. Cellulose fiber is one of the main components of natural fibers found in wood and plants. The composition of cow manure is close to the chemical composition of wood and other plants, so the constituents can be extracted and utilized.

#### 3.1.2. Morphology

The macro form of cleaned cow dung fibers after being washed, disinfected, deodorized, and dried is shown in Figure 2a. It can be seen from the figure that the fibers are light brown, there is no fibrillation on the surface of the fibers, and the fineness of the fibers is nonuniform. In addition, the surface is rough and hard to touch. The morphology of the cow dung fibers was observed by the optical microscope first, as shown in Figure 2b. The cow dung fibers are inhomogenous and consist of tiny fibers and coarse fibers with lengths of <12 mm. The length and width distribution were statistically determined from hundreds of cow dung fibers observed by optical microscopy images (Figure 2c,f). The fiber length ranges from 1–12 mm, and the proportion of each length of fiber are roughly calculated as well. Fibers shorter than 2 mm account for about 15% of the samples whereas the ones longer than 5 mm account for about 33.6%, but in the part of the fibers longer than 5 mm, the length of more than 10 mm is little, about less than 4%. The fiber length is mainly concentrated in the length range of 3~4 mm, accounting for about 51.5%. The width distribution of cow dung is 0.1~1.4 mm, the fiber width less than 0.4 mm accounts for about 17.8%, and greater than 0.9 mm accounts for about 17.5%. Fibers with widths between 0.4 and 0.9 mm accounted for 74.5% of the total. Table 2 shows the distribution statistics of the length and width of cow manure at different scales. The cow dung fibers are not long and crumbled, which may be related to digestion process. Further microstructural analyses of cow dung fibers are illustrated by SEM images of Figure 2d,e. Cow dung fibers have mesoporous structure that further facilitates permeation of liquids (alkaline liquid and hydrogen peroxide solution) during extraction process and increases the cooking efficiency, thus enhancing the extraction rate.

#### 3.1.3. Structure and Thermal Stability Analyses

FTIR analysis of cow dung fibers

FTIR spectra of raw cow dung fibers are shown in Figure 3a. The sample mainly absorbs IR in two regions: from 500 to 1700 cm^−1^ and from 2800 to 3500 cm^−1^. In the spectrum of cow dung fibers, the absorption band at 3310 cm^−1^ is attributed to the O–H stretching vibration of the hydrogen-bonded hydroxyl group of cellulose and lignin [32,33,34,35]. The spectra of samples revealed the aliphatic saturated C-H groups produce a stretch vibration for cellulose and hemicellulose components around 2915 cm^−1^ [33,35]. Cellulose, hemicellulose, and lignin are present in cow dung fibers. The absorption band at 1632 cm^−1^ is assigned to the bending vibration of the -OH group, which may be absorbed water originating from strong cellulose-water interactions [36,37,38]. The absorption peak at 1731 cm^−1^ is attributed to the stretching vibration of C=O bonds of unconjugated ketones and is characteristic of hemicellulose [39,40,41,42]. The band at 1240 cm^−1^ is attributed to the stretching of the acetyl groups of hemicellulose [43] and the one at 1512 cm^−1^ to lignin benzene skeleton stretching vibration [44]. In addition, the peak at 1031 cm^−1^ is associated to the stretching vibration of the pyran ring of cellulose and the one at 896 cm^−1^ to the b-glycosidic linkages between the mono-saccharides and β-glycosidic of cellulose [45]. These peaks are characteristic of the typical cellulose, hemicellulose, lignin, and amorphous structure that existed in the cow dung fibers [46].

XRD analysis of cow dung fibers

XRD was performed on the cleaned-cow dung, as shown in Figure 3b, which presents the XRD measurement for the dried sample of prepared cleaned-cow dung. At present, the XRD diffraction pattern shows significant intensity peaks located at 15°, 21°, and 33°, which correspond to the (110), (200) and (004) planes of cellulose I [14,47,48]. Meanwhile, the relatively high background probably comes from the amorphous glass slide or amorphous structure in cow dung fibers. The degree of crystallinity is about 51.07%.

Thermogravimetric analysis

The thermogravimetric analysis (TG and DTG curves), as illustrated in Figure 3c, was used to characterize the thermal stability of the cow dung fibers. The TG curve showed that the first weight-loss stage was generated from 50 °C to 115 °C [49]. Since fibers are not entirely crystalline, they are capable of absorbing moisture, and the first weight-loss stage is because of the desorption/release of water from the hydrophilic cow dung fiber’s surface. From 115 °C to 200 °C, there is no weight loss or other thermal decomposition. The raw materials showed a novel thermostability and presented a stable stage on the TG curve. Because of the differences in the chemical compositions and structures of hemicellulose, cellulose, and lignin, they usually decompose at different temperatures. Thus, the raw cow dung sample had two decomposition stages. The first occurred at approximately 287 °C and corresponded to the depolymerization of hemicellulose and glycosidic unions [50]. Afterwards, a second decomposition occurred at around 356 °C, which compared to the thermal degradation of cellulose [51]. The high decomposition temperature of cellulose shows the well-ordered structure of cellulose, which is composed of long linear chains of glucose polymers. A broad peak between 200 °C and 400 °C, with a maximum intensity at 360 °C, was observed in DTG curve and attributed to lignin decomposition [51].

### 3.2. Extraction Treatments

#### 3.2.1. The XRD Results and Analyses of the Samples from Different Extraction Treatments

To determine the crystallinity of the samples with different treatments, typical XRD patterns are shown in Figure 4. The diffraction peaks at 2θ = 16°, 22°, and 34° correspond to the (110), (200), and (004) crystal planes of the cellulose I [14,47,48]. Some literature has shown that when the alkali concentration reaches above 10%, the cellulose type changes from I to II [52]. In this study, low-concentration alkali solution (5%) was used to treat cow dung, and cellulose type was only I in the whole experimental process. The structure of the cellulose did not change, which proved that the low concentration of cellulose I fiber extraction is feasible. Compared with the S1, intensity of the crystallinity of the S2, S3, S4, and S5 increased, and S4 and S5 have the highest crystallinity. That is because after the hot water, hydrogen peroxide and alkali boiling treatments, amorphous cellulose was removed and the diffraction peaks intensity became sharp [53].

#### 3.2.2. SEM Results and Analyses of the Samples from Different Extraction Treatments

The SEM analyses of the cow dung fibers by different treatments, as illustrated in Figure 5 evaluate the apparent shape of the crude sample and the modifications after different treatments. In the images corresponding to hot water (Figure 5a–c) and H_2_O_2_ boiling treatments (Figure 5d–f), the samples still have some plant tissue structure and rough surfaces. Regarding samples corresponding to hot water and H_2_O_2_ boiling treatments, cellulose, hemicellulose, lignin, and other water-soluble components are oriented and bonded. The last two samples were, respectively, treated with NaOH (Figure 5g–i) and KOH (Figure 5j–l) boilings; little difference between the smoothness and flatness of the treated samples are observed. After the alkalization step, the cow dung material is de-fibrillated into bundles of fibers. The single bundle of fibers separates, removing the cementitious material (mainly lignin) from the material surface [54]. Alkalization may trigger hydrolysis of some chemical functions (ether and ester bonds) between lignin monomers or between lignin and polysaccharides, which participates to partial defibrillation of the fibers and thus improves the penetration of bleaching solution during bleaching treatment [55]. Therefore, the surface smoothness of alkali-treated samples is smoother than that of hot water and H_2_O_2_ boiling treatments.

#### 3.2.3. FTIR Results and Analyses of the Samples from Different Extraction Treatments

The FTIR patterns of samples S1, S2, S3, S4, and S5 are shown in Figure 6. Two central absorbance regions from 500 to 1700 cm^−1^ and 2800 to 3500 cm^−1^ are observed in all the samples. The absorption bands at 3333 cm^−1^ and 2915 cm^−1^ correspond to the stretching vibrations of -OH and C-H, respectively. Both groups are presented in cellulose, hemicellulose, and lignin, so the corresponding characteristic peaks at 3333 cm^−1^ and 2915 cm^−1^ appear in all samples [43,52]. The peak at 3333 cm^−1^ is more prominent in the FTIR spectra of the S1 compared to the S4 and S5. Such a general decrease of this band from the spectrum of S1 to S5 implies various free OH group amounts. The alkali treatment (S4 and S5) reduces hydrogen bonding because of the removal of the hydroxyl groups by reacting with sodium hydroxide [56,57]. The absorption bands at 1700 cm^−1^ and 1240 cm^−1^ are related to the stretching vibrations of C=O bonds of unconjugated ketones and acetyl groups in hemicellulose [39,43], which is apparent in samples S1 and S2 but almost invisible in samples S4 and S5 after the alkaline treatment. This indicated that the hemicellulose cannot be removed by hot-water treatment but can be effectively be removed by alkaline treatments [58]. In sample S3, weak bands remain at 1700 cm^−1^ and 1240 cm^−1^, demonstrating that hemicellulose cannot be completely removed by the H_2_O_2_ [40]. The absorption peak at 1512 cm^−1^ is attributed to the stretching vibration of the benzene ring carbon skeleton in lignin [44]. The intensity of the absorption peaks of the treated samples S2 is like the one of S1 whereas the one of S3 is reduced. Due to the removal of lignin, the characteristic absorption of hemicellulose at 1700 cm^−1^ is higher in S3 than in S1 and S2. Samples S4 and S5 did not have prominent absorption peaks at 1512 cm^−1^, showing that the alkaline treatments have a better removal effect of lignin. The absorption peaks at 1031 cm^−1^ and 896 cm^−1^, correspond to the stretching vibrations of the pyran ring and of the C-H glycosidic bonds, characteristic of cellulose [46] and as it can be seen from the figure. It can be seen that the peak intensities of samples S4 and S5 are much higher than the other samples while the peak intensity of sample S1 is the lowest, showing that after the alkaline treatments, the hemicellulose, lignin, and amorphous constituents have been removed, revealing that cellulose became the major component [45]. Hence, these FTIR patterns showed that lignin and hemicellulose residuals were almost removed after the alkaline treatment. Therefore, the processing method must be effectively regulated to remove the non-cellulose components.

#### 3.2.4. TGA Results and Analyses of the Samples from Different Extraction Treatments

TG and DTG curves obtained after different treatments and raw jute samples are shown in Figure 7. It was observed that all samples presented a similar behavior. The first thermal event occurred at temperatures below 100 °C for all samples, which corresponds to moisture loss or evaporation of low molecular weight compounds adsorbed during extraction processes. Since fibers are not entirely crystalline, they are capable of absorbing moisture. Table 3 shows that the temperature and sample weight loss of S1, S2, S3, S4, and S5 occurred below 100 °C are 65 °C/10%, 70 °C/4.9%, 65 °C/0.6%, 70 °C/0.8%, and 60 °C/0.6%, respectively. From the data, the fibers with the greatest alpha-cellulose content, the fraction of crystalline cellulose, tend to contain the lowest moisture content, consistently with XRD (Figure 4). As temperature increases, the sample begins to undergo significant thermal decomposition, and a more significant mass loss can be seen in the TG spectra. The data are summarized in Table 3. The initial temperatures (weight loss) of the first obvious thermal decomposition in S1, S2, and S3 samples were 315 °C (13.5%), 322 °C (18.3%), and 300 °C (12.6%), respectively. S4 and S5 did not find obvious decomposition peaks between 210 °C and 315 °C, and the first obvious thermal decomposition temperature (weight loss) of S4 and S5 were 346 °C (39%) and 346 °C (39.7%). The second obvious thermal decomposition temperature (weight loss) of S1, S2, and S3 are 362 °C (25.1%), 370 °C (27.3%), and 362 °C (33.4%), respectively.

Generally, the higher the initial decomposition temperature of the material, the stronger the thermal stability of the material. Hemicellulose does not have a tight molecular structure like cellulose, so its thermal stability is low. The main thermal degradation temperature range of hemicellulose is 210~315 °C [50]. cellulose’s initial degradation temperature is higher (at the temperature range of 310~400 °C) because cellulose is composed of a long string of unbranched glucose polymers. The structure of cellulose is ordered, so its thermal stability is high. Lignin has a complex aromatic ring; its temperature degradation interval covers the entire thermal degradation process, which is the reason why it is difficult to find its thermal decomposition peak in the DTG spectra [50]. The samples S4 and S5 have no obvious decomposition peak between 210 °C and 315 °C. The first decomposition temperature of S4 and S5 are all at 346 °C, higher than S1 (315 °C), S2 (322 °C), and S3 (300 °C). The thermal decomposition temperature (346 °C) of S4 and S5 at 310~400 °C is significantly shifted left compared with S1 (362 °C), S2 (370 °C), S3 (362 °C), indicating that the thermal stability of the sample after alkali treatment is high. It is because of the alkali treatment that effectively removes hemicellulose and some lignin, and samples S4 and S5 are with the highest purity of cellulose. The mass loss in Table 3 shows that between 310 °C and 400 °C, the mass loss of S4 and S5 is greater than that of S1, S2, and S3. It also shows the significant mass loss in the alkali-treated sample because of the decomposition of cellulose.

The thermal stability of raw jute and treated fibers ranged from 260 °C to 300 °C. The sample with the lowest level of thermal stability was the S1 (275 °C). The other samples had greater thermal stability than raw fiber due to the partial or all removal of hemicellulose that was degraded before cellulose. Therefore, because of the removal of the hemicellulose and lignin, Alkaline-treated samples had the highest thermal stability compared to other treatments.

### 3.3. Characterization of Cow Dung Paper

As shown in Figure 8a, the structure of cow dung paper is uniform and the color of the paper is consistent. Further microstructural analyses of cow dung paper by the SEM are shown in Figure 8b,c. From Figure 8b, the cow dung fibers are firmly connected in crisscross in a dense and uniform structure. Figure 8c shows a selected area of cross-section of the cow dung paper, many course fibers can be seen on the surface of the sample, and the cow dung fibers are soft and dispersed, interlaced with each other.

The structural and mechanical properties of the paper made from cow dung was determined and compared to those of paper made from wheat straw, corn stalk, rice straw, and cotton stalk [59] (see Table 4). As the fiber of cow dung paper is more refined than that of other materials, its mechanical properties basically meet the requirements of paper preparation, its fluffiness is higher than that of other agricultural wastes, and its breakage resistance is better than that of cotton stalk. Cow dung paper can be used as filler in the production of paper base film, thus effectively reducing the production cost of paper base film.

## 4. Conclusions

The extraction and application of natural fibers have received much attention due to the environmental problems caused by petroleum-based talents. In this project, cellulose fibers were extracted by four different treatments from cow dung, which has as yet received little attention, and the cellulose fibers with the best properties were used to make handmade paper. The results of the study show that cow dung contains a large amount of cellulose fibers, that are the most efficiently extracted by KOH boiling treatments. The cow dung fibers extracted by KOH were used for papermaking, and the cow dung paper had good mechanical properties (burst index 2.48 KPam^2^/g, tear index 4.83 mNm^2^/g, tensile index 26.72 Nm/g). The black liquor after extraction could be used as fertilizer, making the cow dung cellulose extraction process green and environmentally friendly. In this study, a new pathway is explored for the utilization of cow dung. Cow dung paper can be used as raw material for paper base films in the future for mulch production, reducing soil pollution from plastic mulch. There are many factors affecting the extraction process, and further research can be conducted to explore a more efficient extraction process in terms of solution concentration, extraction time, and temperature.

## Figures and Tables

**Figure 1 materials-16-00648-f001:**
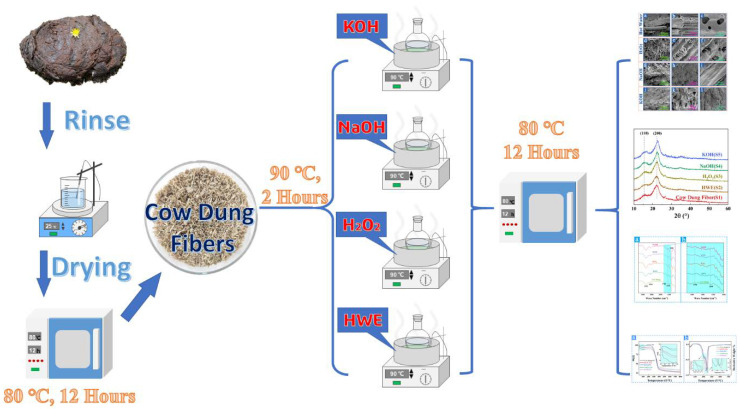
The extraction process of cellulose fibers from cow dung.

**Figure 2 materials-16-00648-f002:**
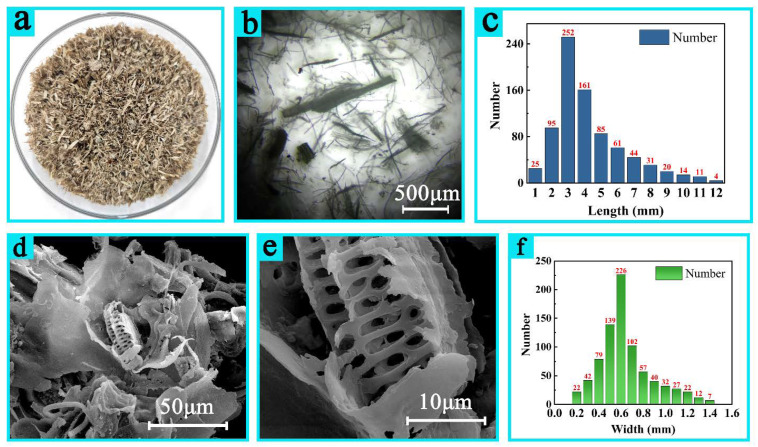
Morphology and length and width distribution of cow dung fibers (macro photo of cow dung after cleaning (**a**); the morphology of cow dung under an optical microscope (**b**); the length distribution of each particle in 1g cow dung (**c**); structure of cow dung under scanning electron microscope (**d**,**e**); the width distribution of each particle in 1 g cow dung (**f**).

**Figure 3 materials-16-00648-f003:**
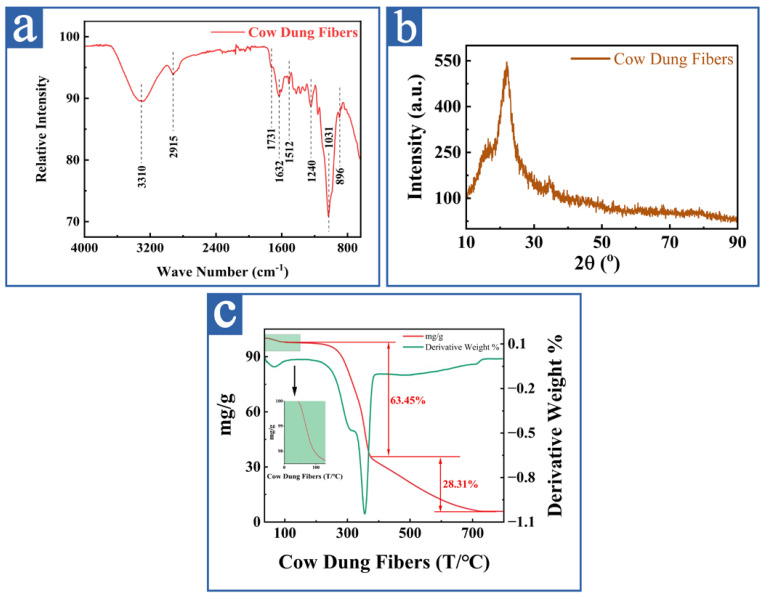
FTIR spectra of raw cow dung fibers (**a**); X-ray diffractogram of cow dung fibers (**b**); and TGA curve of raw cow dung fibers (**c**).

**Figure 4 materials-16-00648-f004:**
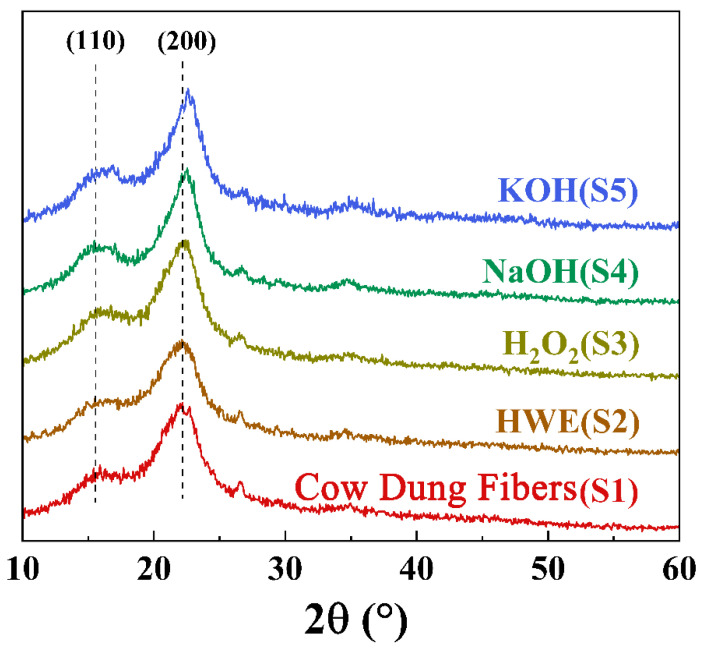
XRD patterns of the samples obtained with different extraction treatments.

**Figure 5 materials-16-00648-f005:**
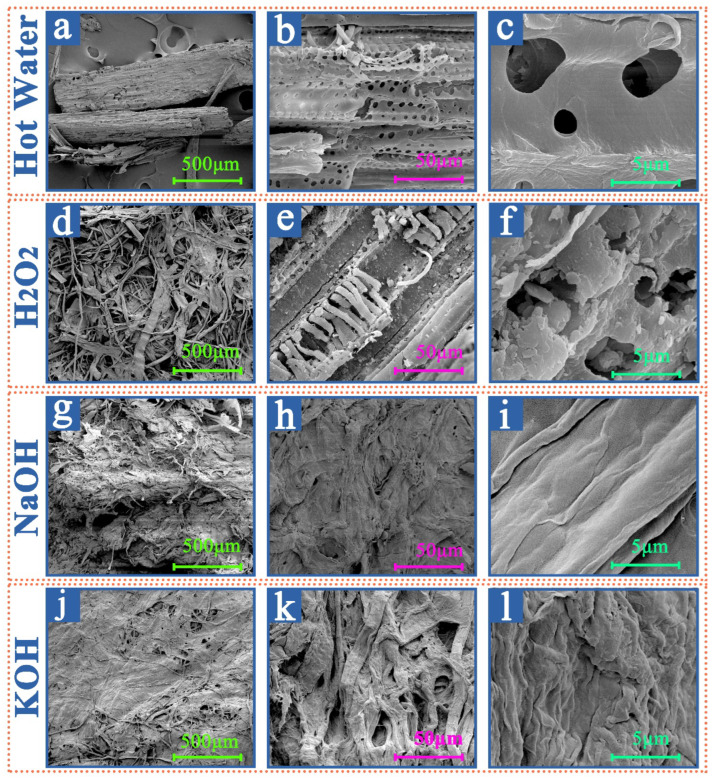
SEM images of the samples from different extraction treatments. (**a**–**c**) The SEM images with different magnification of the samples from Hot water extraction treatment; (**d**–**f**) The SEM images with different magnification of the samples from H_2_O_2_ extraction treatment; (**g**–**i**) The SEM images with different magnification of the samples from NaOH extraction treatment; (**j**–**l**) The SEM images with different magnification of the samples from KOH extraction treatment.

**Figure 6 materials-16-00648-f006:**
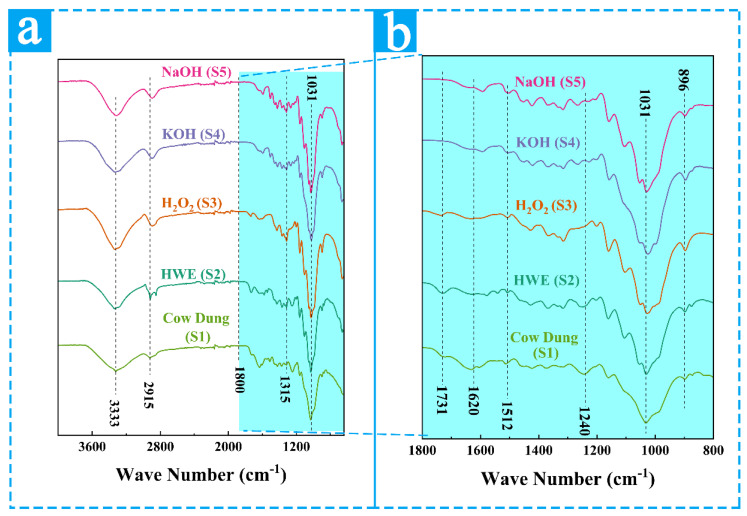
FTIR spectra of the samples from different extraction treatments (**a**) and the enlarge part of the regions from 800 to 1800 cm^−1^ (**b**).

**Figure 7 materials-16-00648-f007:**
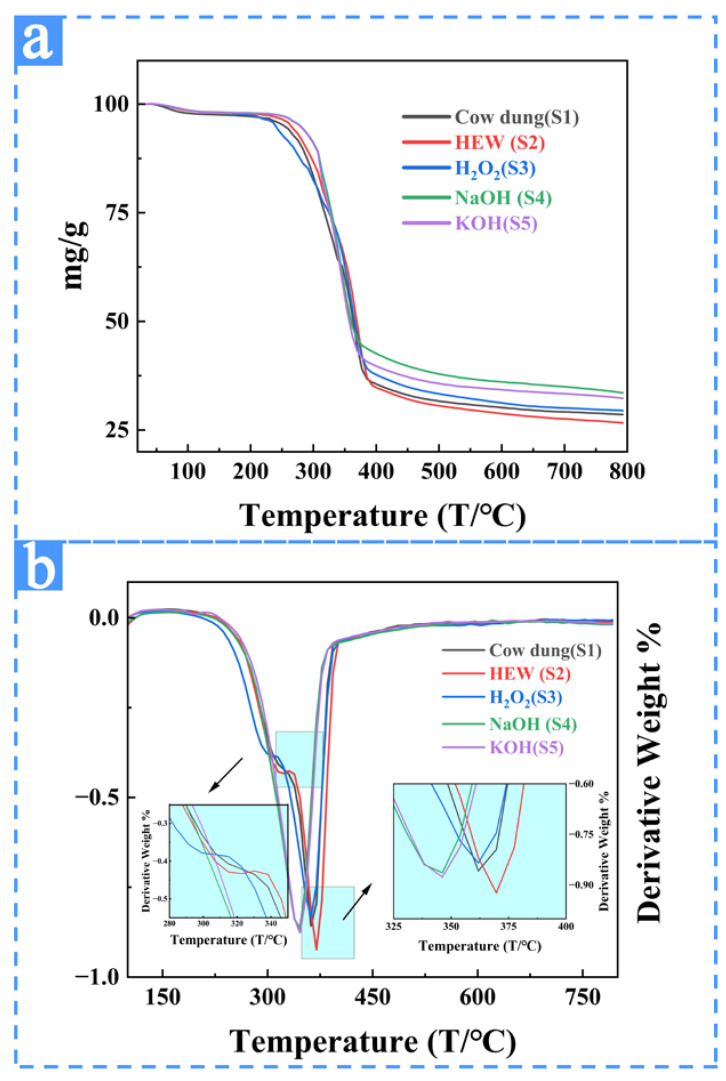
TGA spectra of the samples from different extraction treatments. (**a**) TG spectra of the samples from different extraction treatments; (**b**) DTG spectra of the samples from different extraction treatments.

**Figure 8 materials-16-00648-f008:**
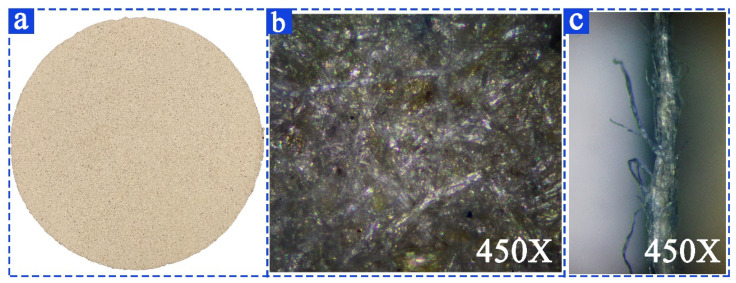
The morphology of cow dung paper. (**a**) The Macroscopic morphology of cow dung paper; (**b**) The SEM image of cow dung paper; (**c**) The SEM image of a selected area of cross-section of the cow dung paper.

**Table 1 materials-16-00648-t001:** Chemical composition of cow dung and other plants.

	CW	HW	NaOH (1%)	OS	Ash	Lign	Hol	Cell
Cow Dung	4.2	4.8	23.7	2.28	5.79	21.31	69.86	41.6
Bagasse [18]	10.6	11.3	15.5		5.0	15.2	64.1	43.0
Cotton stalks [31]		3.33	20.34	1.42	2.17	24.45	14.38	58.48
Softwood [2]	-	-	-	-	-	25–31	65–74	40–45
Hardwood [2]	-	-	-	-	-	16–24	67–82	43–47

CW (wt.%), cold water solubility; HW (wt.%) hot water solubility; OS (wt.%), solubility in various organic solvents; 1% NaOH (wt.%), 1% sodium hydroxide solubility; Hol (wt.%), holocellulose; Lign (wt.%), Klason lignin; Cell (wt.%), cellulose.

**Table 2 materials-16-00648-t002:** The size percentage of the length and width of the cow dung fibers.

**Length (mm)**	**1**	**2**	**3**	**4**	**5**	**6**	**6**	**8**	**9**	**10**	**11**	**12**	**--**
Proportion	3.11%	11.83%	31.28%	20.05%	10.59%	7.6%	5.48%	3.86%	2.49%	1.74%	1.37%	0.5%	--
**Width (mm)**	**0.2**	**0.3**	**0.4**	**0.5**	**0.6**	**0.7**	**0.8**	**0.9**	**1.0**	**1.1**	**1.2**	**1.3**	**1.4**
Proportion	2.76%	5.21%	9.82%	17.18%	27.91%	12.58%	7.06%	4.91%	3.99%	3.37%	2.76%	1.53%	0.92%

**Table 3 materials-16-00648-t003:** Thermal degradation, decomposition peak temperatures, and sample weight loss at different temperatures of the different samples.

Sample	Thermal Stability (°C)	Peak 1	Peak 2	Peak 3	Weight Loss
		$\frac{T_{Peak\;1}\left({}^{\circ}C\right)}{Weight\;Loss\left(\%\right)}$	$\frac{T_{Peak\;2}\left({}^{\circ}C\right)}{Weight\;Loss\left(\%\right)}$	$\frac{T_{Peak\;3}\left({}^{\circ}C\right)}{Weight\;Loss\left(\%\right)}$	100 °C
S1	275 °C	65 °C/10%	315 °C/13.5%	362 °C/25.1%	12
S2	275 °C	70 °C/4.9%	322 °C/18.3%	370 °C/27.3%	6.9
S3	282 °C	65 °C/0.6%	300 °C/12.6%	362 °C/33.4%	1.8
S4	290 °C	70 °C/0.8%	--	346 °C/39%	1.9
S5	292 °C	60 °C/0.6%	--	346 °C/39.7%	1.7

**Table 4 materials-16-00648-t004:** Paper properties made from cow dung or some other sources of agricultural waste.

Raw Materials	°SR (Schopper-Riegler’s Degree)	Grammage (g/m^2^)	Density (kg/m^3^)	Bulkiness (g/cm^3^)	Tear Index (mNm^2^/g)	Burst Index (kPam^2^/g)	Breaking Length (km)	Tensile Index (Nm/g)
Cow dung	42.0	62.5	518..67	1.93	4.83	2.48	2.73	26.72
Wheat straw	57.5	67.2	896.06	1.12	3.58	4.84	7.95	78.00
Corn stalk	41.5	68.68	735.89	1.36	5.36	3.16	5.92	58.01
Rice straw	65.5	69.76	611.02	1.64	4.89	3.42	6.02	59.00
Cotton stalk	45.0	75.40	591.40	1.69	4.75	2.11	3.79	37.13

## Data Availability

Not applicable.

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
