# Peer review of "Characteristics and Functional Application of Cellulose Fibers Extracted from Cow Dung Wastes"

_materials, 2023, doi:10.3390/ma16020648_

Round 1

Reviewer 1 Report

The presented work is related to the extraction of cellulose fibers from cow dung using different treatments and the potential valorization of cow dung to produce mulch. The topic of the work is very interesting. The article is written in an accessible manner, but further details are necessary.

Some grammatical errors and the lack of space between words and references were found and should be re-checked throughout the article. Also, most citations of figures within the whole article gave “Error! Reference source not found” and should be corrected.

Introduction

All the information presented in the introduction is related to the topic of the work. Still, authors should also refer that the interest on the valorization of cow dung is partly due to the continuous growth of the livestock population over the years generating huge livestock wastes, which pose a serious public health issue and environmental pollution; hence the importance of researching alternative but efficient uses of animal waste biomass.

Page 1 – Reference 3 is related to dates from palm tree and not pineapple. When referencing cellulose fibers sources in general, the authors should also mention wood sources (softwood and hardwood).

Page 2 – Change “600,000 heads of castles” by “600,000 heads of cattle”

Experimental procedure

Experimental methods lack some information:

“Cow dung was washed several times and dried”, authors should indicate how many times samples were washed and dried for how long and at which temperature.

Regarding the extraction methods, is there a reason why extraction with hot water was carried out for only 1 hour while extraction with H2O2, NaOH and KOH were carried out for 2 hours? Regarding the extraction conditions, why the authors chose 90°C and 2 hours extraction? Also, information regarding drying time and temperature of extracted samples is necessary.

“Sodium hydroxide extraction (S5)” should be “Potassium hydroxide extraction (S5)”

For FTIR, XRD, TGA and SEM analyses, the model and brand of the equipments, including city and country of origin should be provided.

Regarding FTIR analysis, how were the samples prepared for analysis? Did the authors used KBr pellets or ATR accessory? Were the spectra baseline corrected and normalized? Which normalization technique was used?

Results and discussion

Page 6 – Figure 3 a and b are too small and should be presented separately.

Page 7 – After “…was observed in DTG curve and attributed to lignin decomposition”, Authors should provide a reference for this statement, for instance, Yang et al., Fuel 2007, 86, 1781-1788 (doi:10.1016/j.fuel.2006.12.013).

Page 7 – (Sun et al., 2014) must be formatted as a numbered reference. After “…. Amorphous cellulose was removed and the diffraction peaks intensity became sharp”, authors should provide reference to corroborate the statement.

Page 10 – After “… the hemicellulose cannot be removed by hot water treatment but can be effectively removed by alkaline treatment.”, authors should provide reference to corroborate this statement, for instance Ebringerova and Heinze, Macromol. Rapid Commun. 2000, 21, 542-556 (doi: 10.1002/1521-3927(20000601)21:9<542::AID-MARC542>3.0.CO;2-7)

Page 11 – Regarding TGA results in Figure 7a, between 100 and around 250°C, there is an increase of the mass during the heating experiment. This buoyancy effect, which is due to the change in density of a gas as the temperature changes, should be corrected by performing a blank measurement (baseline) in the exact same conditions (using the same temperature program and crucible) as the experiment but without a sample (empty crucible). The resulting blank curve is then subtracted from the sample measurement curve. Authors should perform this blank experiment to correct the final TGA curves. Was this blank experiment performed? There is no mention in the experimental methods section. If it was, it should be mentioned, if not, to be correct it should be done.

Page 12 – Line 5, After …”and respective weight losses of “, the rest of the sentence is missing. Line 10, Fig. (x) – the figure number is missing.

Page 13 – Authors should provide further discussion on the properties of cow dung paper and compare them with those from papers made from other sources. Furthermore, the potential application for paper-based films for mulch production should also be presented in this section and not only in the conclusions.

References

The authors should verify the DOI numbers of references since many of them are incorrect. Reference 9 is a book and its format is not correct. The title of reference 43 is in uppercase and should be in lowercase. References 40 and 54 are lacking volume, pages and DOI.

Reviewer 2 Report

The manuscript is well organized and the study give a high impact on natural fiber research. it's a new alternative other than plant-based materials.  However, a few things need to be improved as below.

1. for the chemical compositions - what is standard method (need to be clarify for better understanding.)

2. Pls lable the figure 2. and in the figure title - show details what is a,b,c,d

3. why only before treatment does the physical analysis such as diameter and length? pls share the physical properties in the table for a quick review. 

4. Removed some of the old references, better cite paper recent paper (5 years back)

Reviewer 3 Report

The Mancryptus of Xiangjun Yang et al is devoted to the extraordinary problem of obtaining cellulosic waste from cow dung waste.
The authors in their theoretical review explore various examples of non-traditional sources of cellulose and substantiate the choice of research direction.
It is believed that the raw material base contains a pulp mass of about this volume.
For waste treatment and pulp recovery, classical approaches using alkali and H2O2 are used.
The authors estimate the average particle size of cellulose, the content of lignin and other impurities in the polymer and compare the values with those obtained for cellulose from other sources.
Thermal behavior, morphology and structure are evaluated to obtain a sample. Unfortunately, the manuscript contains a large number of typing errors that need to be corrected.
I recommend it to the head of the income discourse department.

“Cellulose fibers can also be recovered from waste biomass such as apple pomace, mulberry leaves, cassava pomace, bagasse, corn stover, peanut shells, etc.
[17,18].

In the manuscript in many cases it occurs - “Error! Reference source not found."
"Extraction with sodium hydroxide (S5)" - needs to be corrected
X-ray diffraction (XRD) - you need to add a powerful method! If this is Segal's method, then why did the authors kill 2θ = 15°?
"Error! Reference source not found. Indicates a chemical" is an unfortunate sentence start.
Table 1. Obscure values obtained when the system was treated with 1% NaOH. How do the authors explain them?
"17.5%.74.5%." - typo
"cow dung" - correct the inscription on the drawings to "cow dung fibers"!
"Chemically absorbed water occurring" - absorbed water is probably common here.
"(#PDF-00-043-1589)" - delete.
“Meanwhile, the relatively high background is likely due to the amorphous slide or amorphous structure in cow dung fibers. However, many other peaks have also been found.
detected that could have been generated by the corresponding silicon dioxide." - remove the warning.
"Figure 4". - fix the inscriptions on the image - "Cow dung".
"Cross-cross" is probably a typo.
Table 3. I recommend the author to describe in more detail the results presented in the table with the above results.

Reviewer 4 Report

The Manuscript on "characteristics and functional application of cellulose fibers extracted from cow dung wastes" is interesting it should need some modifications as the below comments 

1."Four different treatments (Hot water, H2O2, NaOH, and Sodium KOH boiling extractions) were used to extract cellulose fibers from crude fiber, and the specific treatment process is shown in Error! Reference source not found.." no need to mention the error.

2.Figure-1, need more specificity like which types of treatment are mentioned near the arrows, temperature, acid...etc.

3. Figure 2 and all other figures.  a,b,c,d,...........need captions of images. 

4. The macro form of cleaned cow dung fibers after being washed, disinfected, deodorized, and dried is shown in Error! Reference source not found.a. why are authors mentioning errors? from throughout the manuscript it should remove if those your explanations.

5.Top of Figure 6 : " FTIR spectra of the samples from different extraction treatments:" this line wrote should remove.

6. °SR in Table 3, should mention the full form below the table as a note. 

7. Please provide BET for every sample, surface, and pore size. 

Round 2

Reviewer 4 Report

The Manuscript on "characteristics and functional application of cellulose fibers extracted from cow dung wastes". The problems are compiled by the authors very carefully. 

If you add some recent and appropite references in your FTIR and XRD results, It will be more attractive. Like DOI: 10.1039/C9BM01341EDOI: 10.1039/C5RA25402G
